# Endoscope-Assisted Extreme Lateral Supracerebellar Infratentorial Approach for Resection of Superior Cerebellar Peduncle Pilocytic Astrocytoma: Technical Note

**DOI:** 10.3390/children9050640

**Published:** 2022-04-29

**Authors:** Kyriakos Papadimitriou, Giulia Cossu, Ekkehard Hewer, Manuel Diezi, Roy Thomas Daniel, Mahmoud Messerer

**Affiliations:** 1Department of Neurosurgery, University Hospital of Lausanne and University of Lausanne, 1015 Lausanne, Switzerland; kpapademetriou1@gmail.com (K.P.); giulia.cossu@chuv.ch (G.C.); roy.daniel@chuv.ch (R.T.D.); 2Department of Pathology, University Hospital of Lausanne and University of Lausanne, 1015 Lausanne, Switzerland; ekkehard.hewer@chuv.ch; 3Hemato-Oncology Unit, Department of Pediatric, University Hospital of Lausanne and University of Lausanne, 1015 Lausanne, Switzerland; manuel.diezi@chuv.ch

**Keywords:** superior cerebellar peduncle, extreme lateral supracerebellar infratentorial approach, glioma, pilocytic astrocytoma

## Abstract

(1) Background: Superior cerebellar peduncle (SCP) lesions are sparsely reported in the literature. The surgical approaches to the cerebello-mesencephalic region remain challenging. In this article, we present the extreme lateral supracerebellar infratentorial (ELSI) approach to treat a large hemorrhagic pilocytic astrocytoma of the SCP. (2) Methods: An 11-year-old boy, known for neurofibromatosis Type I, presented to the emergency department of our institution with symptoms and signs of intracranial hypertension. The cerebral magnetic resonance imaging (MRI) revealed a large hemorrhagic lesion centered on the SCP provoking obstructive hydrocephalus. Following an emergency endoscopic third ventriculocisternostomy (ETV), he underwent a tumor resection via an endoscope-assisted ELSI approach. (3) Results: ELSI approach allows for a wide exposure with direct access to lesions of the SCP. The post-operative course was uneventful, and the patient was discharged home on post-operative day 5. Post-operative MRI revealed a near total resection with a small residual tumor within the mesencephalon. (4) Conclusion: ELSI approach offers an excellent exposure with the surgical angles necessary for median and paramedian lesions. The park-bench position with appropriate head flexion and rotation offers a gravity-assisted relaxation of the tentorial and petrosal cerebellar surfaces. The endoscope can be an adjunct to illuminate the blind areas of the surgical corridor for an improved tumor resection without significant cerebellar retraction.

## 1. Introduction

Pilocytic astrocytomas (PAs) are the most frequent primary intracranial tumor in children and the majority of them arise from the cerebellum. Their optimal management includes gross-total resection (GTR), which is associated with an excellent progression free survival [1]. However, in case of peduncular involvement, GTR may be challenging [2], and in these cases, maximal safe resection with preservation of neurological function is recommended [3]. Spontaneous intratumoral hemorrhage is rare with PAs and its etiology is likely to be due to abnormal tumor vasculature [4,5].

The paramedian variant of the supracerebellar infratentorial approach was described by Yasargil and Voigt for the resection of a cavernous malformation of the parahippocampal gyrus [1,6]. The extreme lateral supracerebellar infratentorial (ELSI) approach was then described by Vishteh et al. to access the middle and posterolateral incisural space, which corresponds to the ambient and posterolateral aspect of the quadrigeminal cistern [7]. ELSI was then popularized by the same group to address brainstem cavernous malformation [8].

Nowadays, ELSI is considered a versatile approach that is used to access pathologies in the posterior incisura space and cerebello-mesencephalic fissure, namely in pineal region, posterior thalami, dorsal midbrain and pons, quadrigeminal and ambient cisterns [8,9,10]. Opening the thick arachnoid membrane of the quadrigeminal cistern and of the cerebello-mesencephalic fissure provides a safe and natural corridor to the posterior incisura space while preserving the critical neurovascular structures, such as the Galenic venous complex, the posterior cerebral artery (PCA), the pineal gland, the posterior wall of the midbrain including the upper part of the collicular plate, the posterior wall of the third ventricle and the superior cerebellar peduncle (SCP) [11].

In this paper, we present our experience using endoscope-assisted ELSI approach to resect a large hemorrhagic PA of the SCP.

## 2. Case Presentation

An 11-year-old boy, known for Neurofibromatosis (NF) Type I presented to the emergency department of our institution with symptoms and signs of intracranial hypertension. On neurological examination, he presented drowsiness and a partial left VI cranial nerve palsy. A cerebral magnetic resonance imaging (MRI) revealed a large hemorrhagic lesion centered on the left SCP associated with an obstruction of the aqueduct of Sylvius and resultant obstructive hydrocephalus (Figure 1). The differential diagnosis was pilocytic astrocytoma, medulloblastoma, germ cell tumor, atypical teratoid/rhabdoid tumor. In this context, he underwent an emergency endoscopic third ventriculocisternostomy (ETV). The postoperative evolution was favorable with partial regression of VI cranial nerve palsy. Cerebrospinal fluid (CSF) analysis for alpha fetoprotein, b-hCG and placental alkaline phosphatase was negative. The three weeks post-ETV MRI showed regression of the hydrocephalus and a resorption of the intra-tumoral hematoma (Figure 2).

## 3. Operative Technique 

(Appendix A). Following general anesthesia and orotracheal intubation the patient was positioned in right lateral decubitus position, with the head immobilized in a Mayfield head-clamp slightly flexed and turned 45° towards the floor in order to place the genu of the sigmoid sinus at the highest point of a perpendicular line that bisects the sinodural angle (Figure 3). Reverse Trendelenburg position is applied to the surgical bed to facilitate venous return. This position allows a “double relaxation” of the petrosal and tentorial cerebellar surfaces. Moreover, this maneuver will allow panoramic vision with the operative microscope to facilitate the access to the supracerebellar window and retrosigmoid cerebello-pontine angle as well as in the depth to the ambient and quadrigeminal cisterns.

A hockey-stick incision extending from the inion to the spinous process of C2 was performed. The suboccipital muscles are dissected layer by layer using monopolar electrocautery. The atlanto-occipital membrane is opened with care utilizing micro-scissors. The foramen magnum is afterwards opened using Kerrison punch. A suboccipital craniotomy with a lateral extension (extreme lateral) is performed. The entire left transverse sinus, the transverse-sigmoid junction, and the proximal part of the sigmoid sinus are exposed.

A small durotomy is performed at the inferior edge of the craniotomy and cerebro-spinal fluid (CSF) is aspirated from the lateral cerebello-medullary cistern to relax the cerebellum. The dura is then widely opened about 5–8 mm inferior to the transverse sinus and medial to the sigmoid sinus in a curvilinear fashion. Three dura retraction sutures are placed below the transverse sinus, at the transverse-sigmoid junction, and medial to the sigmoid sinus. Arachnoid dissection is performed to free the tentorial surface of the cerebellum. Bridging veins, including the precentral cerebellar and superior vermian vein are preserved if they do not interfere with the surgical corridor. On the tentorial surface of the cerebellum three fissures are encountered in the way towards the SCP and can be used as anatomical landmarks. Superficial to deep they are the preculminate sulcus (between the culmen and central lobule), precentral cerebellar sulcus (between the central lobule and the lingula and the fissure between the lingula and the SCP. It is important to emphasize that the lingula lays on the dorsal surface of the SCP sometimes without clear demarcation (Figure 4) [11]. The next step is the identification of the arachnoid plane of the internal cerebral veins and the tumor in the quadrigeminal cistern, which is sharply dissected. Following identification of the SCP and the vascular structures such as the Galenic venous complex, internal cerebral veins, veins of Rosenthal and PCA that are located on the superior pole of the tumor, tumor resection is started from its exophytic part.

Tumor resection is then continued with CUSA and aspiration in a piece-meal fashion. The intra-tumoral resection is proceeded while respecting the pial planes laterally. If the cerebellar relaxation is not adequate, a posterior ventriculostomy can be performed by opening the posterior wall of the third ventricle between the splenium and the pineal gland. The resection is done progressively under the operating microscope; however, limited vision is encountered on the superior and inferior angles of the surgical corridor. To improve tumor visualization, the 30° endoscope may be introduced along the surgical corridor to continue resection under direct vision and maximize tumor resection. Following placement of the operating endoscope, tumor remnants may be identified on the superior portion of the surgical corridor, and they are carefully resected by small, angulated suctions. A small part of the tumor was left in place, because of its attachment to the mesencephalic part of the SCP. After meticulous hemostasis, the dura was closed in a water-tight fashion followed by replacement of the bone flap. The skin closure was performed in layers.

The post-operative course was uneventful, with no new neurological deficits. Immediate post-operative MRI showed a near total resection with a small residual tumor within the mesencephalon (Figure 5). The histological examination revealed a PA (Figure 6). The patient was discharged home on post-operative day 5.

## 4. Discussion

Pediatric low-grade gliomas encompass a heterogeneous group of tumors and PA is the most common brain tumor in the first two decades of life [3]. They can be found along the neuraxis, including the optic chiasm, hypothalamus, cerebral hemisphere, brainstem and cerebellum. Among them, cerebellar Pas is the most frequently encountered localization [12] but peduncular involvement remains rare [13].

Complete surgical resection should be attempted if possible, when a gross total resection (GTR) is achieved, no further treatment is deemed necessary [1,14]. GTR is associated with a 5-year survival rate of 95% and it offers progression free survival between 71% and 100% of cases [1,15]. However, Pas arising from the optic pathways, hypothalamus, brainstem and cerebellar peduncles are usually not amenable to GTR. Recent advances in neuro-imaging and micro neurosurgery allow near total resection (NTR) or even GTR of such lesions with favorable outcomes [16,17]; however, the preservation of neurological functions remains a primary goal of the treatment. Residual tumors can be safely managed with adjunct treatments, such as targeted chemotherapy based on tumor’s molecular genetics and radiotherapy [13].

Hemorrhagic presentation of Pas is rare, and it is observed more commonly in adults and few anecdotal reports in children till date [14,15]. Several theories have been proposed for the occurrence of intratumoral hemorrhage in PA and includes abnormal vasculature such as thin-walled ectatic blood vessels, degenerative mural hyalinization and glomeruloid endothelial hyperplasia [5]. Another key factor is the higher VEGF expression in PA. Some authors have noted that vascular integrity in cerebellar Pas is unstable in a similar fashion that of glioblastomas [18]. Moreover, NF-1 is associated with cerebrovascular diseases, pathologic vasculature, and moyamoya disease, and therefore the incidence of intratumoral hemorrhage in NF-1 patients might differ from that of sporadic PA [14]. Hemorrhagic PA are associated with a higher percentage of morbidity and mortality than non-hemorrhagic Pas, due to intracranial hypertension [15]. Moreover, pathological analysis of hemorrhagic Pas did not reveal more aggressive features when compared to non-hemorrhagic Pas [19].

SCP, or brachium conjunctivum, is a paired bundles of white matter fibers that connects the cerebellum to the midbrain and it is associated with coordination of muscle activity and cognitive function (Figure 7). The SCP is divided into three segments and named from a surgical perspective: (1) initial or congregated, (2) intermediate or intraventricular and (3) distal or intramesencephalic segments [20] (Figure 7). The congregated segment leaves the hilum of the dentate nucleus and reaches the lateral border of the upper half of the fourth ventricle, where it is called the ventricular segment, joined by the lateral lemniscus in the cerebello-mesencephalic fissure. The ventricular segment is in close proximity to the superior medullary velum medially within the cerebello-mesencephalic fissure in the quadrigeminal cistern. As the SCP fibers climb superiorly to its junction with the tectum, the ventricular segment is located deep to the inferior colliculus to become the distal intramesencephalic segment.

The SCP represents the main outflow of the cerebellum, and it contains efferent fibers from the dentate nucleus, medial emboliform, and globose nuclei to the contralateral ventrolateral thalamus and red nucleus [21]. As these fibers emerge from the hilum of the dentate nucleus with an ascending trajectory (ascending division), they form a compact bundle along the dorsolateral surface of the fourth ventricle. As they sweep supero-medially into the tegmentum, they decussate at the level of inferior colliculus behind the medial lemniscus [21]. A small percentage of these crossed fibers arising from the dentate nucleus terminate at the level of the rostral third of the red nucleus. On the other hand, fibers from the emboliform and globose nuclei are projecting to the caudal two thirds of the red nucleus. The vast majority of the crossed fibers from all cerebellar nuclei surround the red nucleus and continue rostrally to terminate in the ventral lateral (VL) and ventral posterolateral thalamic (VPL) nuclei [21]. Thereafter, VL and VPL project to the primary sensory cortex. Notably, these fibers that terminate in the thalamic nuclei present a somatotopic organization, with the head is represented medially, the extremities lie ventrally, and the back lies dorsally. This circuit is associated with coordination of somatic motor function. Lastly, some other fibers from the dentate nucleus project via the SCP to the intralaminal thalamic nuclei and specifically the central lateral nucleus. A smaller part of fibers following the SCP decussation, descend (descending division) to the reticulotegmental nucleus and inferior olive. Some other fibers of the descending division project back to cerebellar cortex of the hemisphere (double decussation) [21]. (Figure 7).

It is worth mentioning that fastigial efferent fibers, although they do not emerge via the SCP, form the uncinate fasciculus of Russell around the SCP as they cross to the contralateral side, to reach the vestibular nuclei. Another group of fibers of the uncinate fasciculus, ascend ipsilaterally to terminate according to a somatotopic distribution in the VL et VPL. A smaller group of fibers instead, descend to the ipsilateral side to end in the vestibular nuclei and reticular formation [21].

Keeping in consideration these anatomical and functional connections of the deep cerebellar nuclei via the SCP, lesions of the SCP may result in variable clinical symptoms, as listed before [21]. 

The clinical course of SCP injury remains peculiar. Kim et al., investigated the state of the SCP using diffusion tensor tractography (DTI) in a 6-year-old girl treated surgically for PA of the left cerebellar hemisphere [22]. Pre-operatively, she had severe ataxia, with a score of 0 separately on the Berg’s Balance Scales. However, at the 3-month follow-up, the results of DTI showed increased fractional anisotropy values of both SCPs with values comparable to healthy controls. The authors assumed that SCPs fibers may recover after injury, as shown by the clinical course of the patient.

From a surgical standpoint, the SCP is located near the cerebello-mesencephalic sulcus at the dorsolateral midbrain, corresponding to the junction of the quadrigeminal and ambient cisterns. Anatomical relationships in this region consists anteriorly in the pineal body rostrally, the quadrigeminal plate, the trochlear nerve and the lingula of the vermis on the midline [23]. It is related superiorly to the lower surface of the splenium, the crura of the fornices, and the hippocampal commissure, and laterally to the pulvinar, the crus of the fornix, the medial surface of the posterior part of the parahippocampal and dentate gyri. The arterial relationships consist of the PCA and superior cerebellar artery (SCA). The venous relationships are with the two internal cerebral veins and the basal veins that exit the velum interpositum and the ambient cistern, respectively, to reach the posterior incisural space, where they join to form the vein of Galen [23].

Traditionally, the supracerebellar infratentorial approach, with it is a variant of the median or paramedian approach, have been utilized for lesions in the cerebello-mesencephalic fissure and posterior incisural space, including pathologies localized on the SCP. However, other approaches have been reported in the literature, such as the occipital transtentorial, and ELSI approach [8,9,10,11,24]. The ELSI approach allows wide exposure of the middle and posterolateral incisural spaces, which are centered on the lateral and posterolateral surface of the mesencephalic tegmentum, respectively [8]. The ELSI was chosen in this case as it is associated with less retraction of the cerebellum as it slopes downward from medial to lateral and enables a more inferior exposure [8]. The amount of bridging veins needed to be sacrificed is less compared to the other variants, thus lowering the risk of venous infarction and hemorrhagic complications [8,9]. The park-bench position was chosen in order to avoid the known complications of the alternative position, namely the semi-sitting position with its risks of air embolism, intracranial hypotension and tension pneumocephalus. Moreover, the park-bench position, with the head flexed and turned 45o towards the floor, offers gravity-aided relaxation of the tentorial and petrosal surfaces of the cerebellum.

Endoscopic assistance in microsurgical procedures was first reported by Hopf and Perneczky as an adjunct to the microscopic manipulations [25]. Beltagy et al. reported their experience in endoscopic-assisted microsurgery for various pathologies including posterior fossa tumors in pediatric population [26]. The authors concluded that endoscopic assistance provides a panoramic view and excellent illumination for a safe assessment of the tumor’s relationship to critical structures with minimal amount of cerebellar retraction [26]. In our case, the 30° endoscope allowed the visualization tumor extensions to the hidden corners of the surgical corridor without any significant cerebellar retraction, allowing a near total resection.

## 5. Conclusions

The ELSI approach offers a wide exposure with multiple surgical angles for median and paramedian lesions located in the posterior incisural space arising from the superior cerebellar peduncle without placing at risk critical neurovascular structures. ELSI in park-bench position with appropriate head flexion and rotation offers a gravity-aided relaxation of the tentorial and petrosal cerebellar surfaces. Additionally, posterior third ventriculostomy can be safely performed to obtain additional brain relaxation. The endoscope is an adjunct to illuminate the blind areas of the surgical corridor to improve tumor resection without significant cerebellar retraction.

## Figures and Tables

**Figure 1 children-09-00640-f001:**
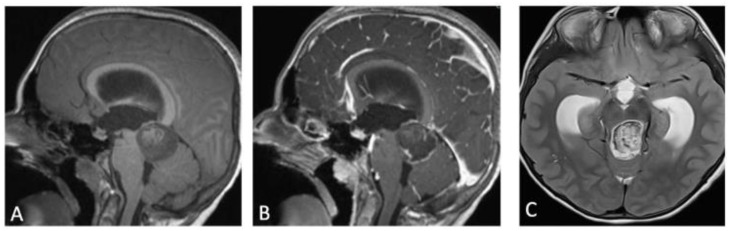
(**A**) T1-weighted MRI sagittal image showing a mass centered on superior cerebellar peduncle, spontaneously iso-intense. (**B**) Contrast enhanced T1 sagittal MRI showing peripheral enhancement. (**C**) T2 axial MRI revealing a heterogeneous signal intensity suggesting acute intralesional hemorrhage and obstructive hydrocephalus.

**Figure 2 children-09-00640-f002:**
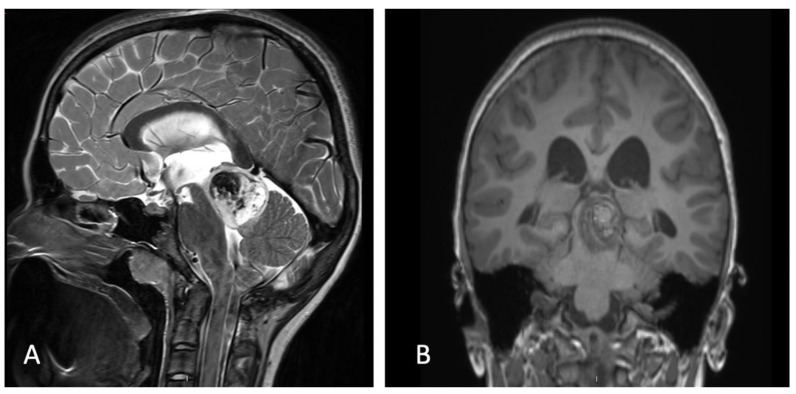
(**A**) Post-ETV CISS MRI showing patent stoma and regression of hydrocephalus. (**B**) Coronal T1-weighted MRI showing a well localized mass on the SCP.

**Figure 3 children-09-00640-f003:**
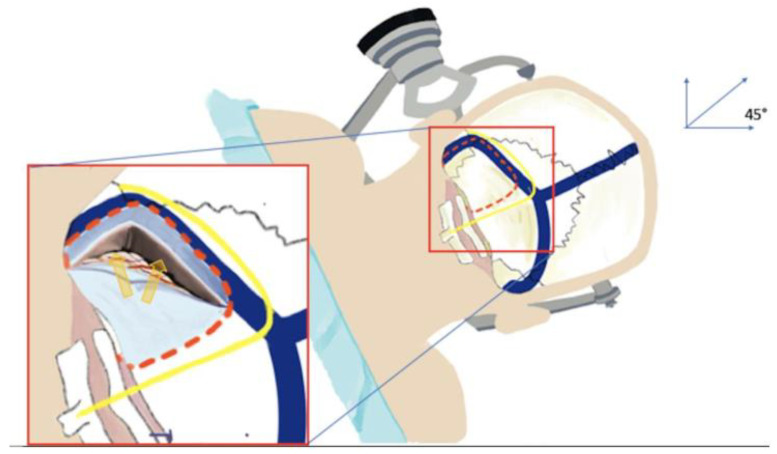
Artist work showing the appropriate head positioning. The head is slightly flexed and turned 45° towards the floor in order to place the genu of the sigmoid sinus at the highest point of a perpendicular line that bisects the sinodural angle.

**Figure 4 children-09-00640-f004:**
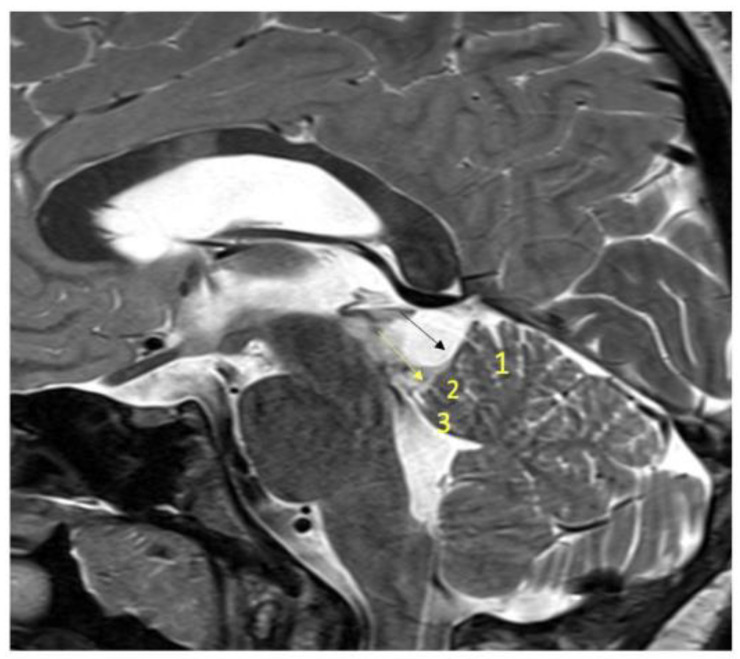
T2-weighted MRI showing the following:(1:)culmen; (2) central lobule; (3) lingula; black arrow: the preculminate sulcus between the culmen and central lobule; yellow arrow: precentral cerebellar sulcus between the central lobule and the lingula.

**Figure 5 children-09-00640-f005:**
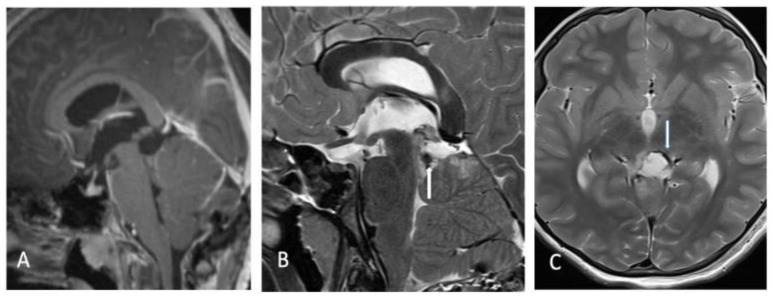
Immediate post-operative MRI (**A**) sagittal contrast enhanced, (**B**) T2 sagittal and (**C**) T2 axial showing near complete resection with a residual tumor on the latero-inferior surface of the tectum (arrow).

**Figure 6 children-09-00640-f006:**
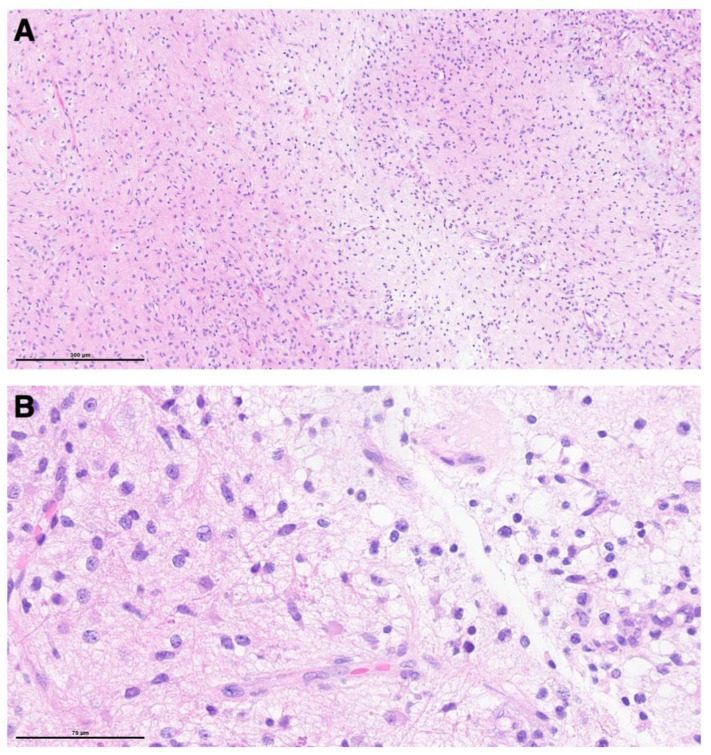
Histological analysis showed a moderately cellular glial neoplasm with biphasic growth pattern (**A**), characterized by tumor cells with bipolar processes and the presence of occasional eosinophilic granular bodies (**B**).

**Figure 7 children-09-00640-f007:**
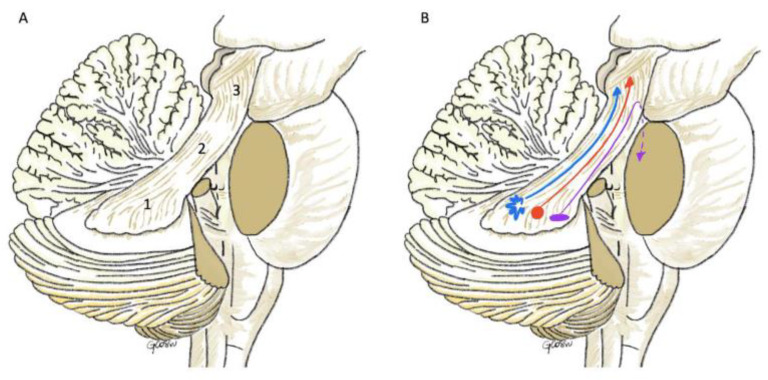
Artist work showing the anatomy of the superior cerebellar peduncle and its connection with the red nucleus and thalamus. (**A**): (1) Initial or congregated, (2) intermediate or intraventricular and (3) distal or intramesencephalic segments. (**B**): The blue line represents the fibers that terminate in the VL and VPL nuclei, the red line the fibers that terminate in the red nucleus and the purple line the fibers of the descending division towards the cerebellar cortex.

## Data Availability

Not applicable.

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
