# Peer review of "Endoscope-Assisted Extreme Lateral Supracerebellar Infratentorial Approach for Resection of Superior Cerebellar Peduncle Pilocytic Astrocytoma: Technical Note"

_children, 2022, doi:10.3390/children9050640_

Round 1

Reviewer 1 Report

The author should continue his researches on the relation of operational access and post-surgical deficites. The prest

Author Response

Reviewer #1:

The author should continue his researches on the relation of operational access and post-surgical deficites. 

Response:

We thank you for the comment. Definitely, this subject needs further investigation.

Reviewer 2 Report

In this article, Papadimitriou K et al. reported the case of pilocytic astrocytoma located at superior cerebellar peduncle presenting with intratumoral hemorrage and surgically removed via extreme lateral supracerebellar infratentorial approach with endoscopic assistance.

This case may serve as a reference to the readers including not only neurosurgeons but also radiologists, oncologists, pathologists and pediatricians.

I have some comments.

  1. First of all, the tumor was originated from the superior cerebellar peduncle (Figure 4)? Please show the post-ETV MRI showing the anatomical limits of the tumor well defined in the SCP (page 2, line 56-58).
  2. It would be better to show the anatomical landmarks in operation Video 1 (supplementary material).
  3. Please show the photomicrographs of histological examination of the tumor (page 3, line 115).
  4. In Figure 5, what are “1”, “2”, “3” in A and blue, red, purple arrows in B? More detail figure legend would be necessary.
  5. Is there any literature describing the peduncular involvement of cerebellar tumors including pilocytic astrocytoma? What kind of preoperative differential diagnoses could be there?
  6. How many cases of pilocytic astrocytoma presenting with intratumoral hemorrhage in the literature?
  7. Recently, “exoscopic neurosurgery” has become popular. How about “exoscopic surgery” in this case?

Author Response

First of all, the tumor was originated from the superior cerebellar peduncle (Figure 4)? Please show the post-ETV MRI showing the anatomical limits of the tumor well defined in the SCP (page 2, line 56-58).

Response:

The tumor was well defined on the SCP. We have added the post-ETV MRI (Figure 2).

It would be better to show the anatomical landmarks in operation Video 1 (supplementary material).

Response:

We have made the appropriate modifications on Video 1.

Please show the photomicrographs of histological examination of the tumor (page 3, line 115).

Response:

We have added the histopathological images (Figure 5).

In Figure 5, what are “1”, “2”, “3” in A and blue, red, purple arrows in B? More detail figure legend would be necessary.

Response:

Figure 5 has been named as figure 6. Artist work showing the anatomy of the superior cerebellar peduncle and its connection with the red nucleus and thalamus. (1) initial or congregated, (2) intermediate or intraventricular and (3) distal or intramesencephalic segments. The blue line represents the fibers that terminate in the VL and VPL nuclei, the red line the fibers that terminate in the red nucleus and the purple line the fibers of the descending division towards the cerebellar cortex. We have made the appropriate clarifications on Figure 5.

Is there any literature describing the peduncular involvement of cerebellar tumors including pilocytic astrocytoma?

Response:

Yes, there are a few papers describing the peduncular involvement.

We added in the text section discussion: “Pediatric low-grade gliomas encompass a heterogeneous group of tumors and PA is the most common brain tumor in the first two decades of life. They can be found along the neuraxis, including the optic chiasm, hypothalamus, cerebral hemisphere, brainstem and cerebellum. Among them, cerebellar PAs is the most frequently encountered localization  but peduncular involvement remains rare.”

What kind of preoperative differential diagnoses could be there? We added in the section Case presentation:

Response:

“The preoperative differential diagnosis is pilocytic astrocytoma, medulloblastoma, germ cell tumor, and atypical teratoid/rhabdoid tumor. We have made the appropriate corrections in the manuscript and cited the appropriate papers”.

How many cases of pilocytic astrocytoma presenting with intratumoral hemorrhage in the literature?

Response: We added this paragraph to the discussion

“Hemorrhagic presentation of PAs is rare, and it is observed more commonly in adults and few anecdotal reports in children till date . Several theories have been proposed for the occurrence of intratumoral hemorrhage in PA and includes abnormal vasculature such as thin-walled ectatic blood vessels, degenerative mural hyalinization and glomeruloid endothelial hyperplasia. Another key factor is the higher VEGF expression in PA. Some authors have noted that vascular integrity in cerebellar PAs is unstable in a similar fashion that of glioblastomas. Moreover, NF-1 is associated with cerebrovascular diseases, pathologic vasculature, moyamoya disease and therefore the incidence of intratumoral hemorrhage in NF-1 patients might differ from that of sporadic PA. Hemorrhagic PA are associated with higher percentage of morbidity and mortality than nonhemorrhagic PAs due to intracranial hypertension. Moreover, pathological analysis of hemorrhagic PAs did not reveal more aggressive features when compared to non-hemorrhagic PAs. “

Recently, “exoscopic neurosurgery” has become popular. How about “exoscopic surgery” in this case?

Response:

We do not have any experience utilizing the exoscope in this kind of tumor surgery and therefore we are unable to comment on this point.

Reviewer 3 Report

Introduction is succinct, would be good to briefly describe role of surgery in pilocytic astrocytoma (with GTR, no further treatment may be required which comes with side effects). 

In case presentation would be more clear regarding tumor markers in CSF.  Presume authors are referring to aFP and bHCG, however with CSF liquid biopsy approaches becoming more widely used would be good to have clarity. 

Operative technique very clear.  Illustrations nicely augment the text. 

Figure 3 appears to describe numbered structures in the image, however no numbers are visible in the image.  Also line 123 "cere-bellar" is hyphenated but on the same line.

In discussion, again would be good to briefly mention role and importance of surgical resection in PA and impact on prognosis and further management.  Chemotherapy and molecular targets may be spared with a good resection. 

Line 153, should this read "as they sweep ventromedially...?

Line 205, should this ready "bridging veins needed to be sacrificed is less compared to the other variants..."

Overall well written and very clearly described surgical approach in this technical note. 

Author Response

Reviewer #3 :

Introduction is succinct, would be good to briefly describe role of surgery in pilocytic astrocytoma (with GTR, no further treatment may be required which comes with side effects). 

Response:

We added the following paragraph.

Pilocytic astrocytomas (PAs) are the most frequent primary intracranial tumor in children and the majority of them arise from the cerebellum. Their optimal management includes gross-total resection (GTR), which is associated with an excellent progression free survival. However, in case of penducular involvement, GTR may be challenging and in these cases maximal safe resection with preservation of neurological function is recommended.

In case presentation would be more clear regarding tumor markers in CSF.  Presume authors are referring to aFP and bHCG, however with CSF liquid biopsy approaches becoming more widely used would be good to have clarity. 

Response:

The tumor markers that we are referring were the aFP, bHCG and placental alkaline phosphatase. We made the appropriate modifications.

Operative technique very clear.  Illustrations nicely augment the text. 

Response:

We appreciate your kind comment.

Figure 3 appears to describe numbered structures in the image, however no numbers are visible in the image.  Also line 123 "cere-bellar" is hyphenated but on the same line.

Response:

In Figure 3 illustrates: 1: culmen, 2: central lobule, 3: lingula, black arrow: the preculminate sulcus between the culmen and central lobule and yellow arrow: precentral cerebellar sulcus between the central lobule and the lingula.

In discussion, again would be good to briefly mention role and importance of surgical resection in PA and impact on prognosis and further management.  Chemotherapy and molecular targets may be spared with a good resection. 

Response:

We have added a paragraph in the discussion section outlining the importance of gross total resection.

Complete surgical resection should be attempted if possible, when a gross total resection (GTR) is achieved, no further treatment is deemed necessary.  GTR is associated with a 5-year survival rate of 95% and it offers a good progression free survivalRecurrence after GTR is reported between 0% and 29%. However, PAs arising from the optic pathways, hypothalamus, brainstem and cerebellar peduncles are usually not amenable to GTR. Recent advances in neuro-imaging and microneurosurgery allow near total resection (NTR) or even GTR of such lesions with favorable outcomes ; But the preservation of neurological functions remains a primary goal of the treatment. Residual tumors can be safely managed with adjunct treatments, such as targeted chemotherapy based on tumor’s molecular genetics and radiotherapy.

Line 153, should this read "as they sweep ventromedially...?

Response:

The correct wording is supero-medially. We made the appropriate modification.

Line 205, should this ready "bridging veins needed to be sacrificed is less compared to the other variants..."

Response:

Yes, this phrase is correct. The extreme lateral supracerebellar infratentorial approach requires less amount of bridging veins coagulation comparing to median or paramedian supracerebellar infratentorial approach.

Overall well written and very clearly described surgical approach in this technical note. 

Response:

We thank you for this comment.

Round 2

Reviewer 2 Report

The authors responded to the reviewer's comments.

Author Response

We thank you for your comment.